# Measurement of Quasi-Static and Dynamic Displacements of Footbridges Using the Composite Instrument of a Smartstation and an Accelerometer: Case Studies

**Jiayong Yu** [1,*] **, Zhen Fang** [1] **, Xiaolin Meng** [1,2] **, Yilin Xie** [3] **and Qian Fan** [4]

[1]   Key Laboratory for Wind and Bridge Engineering of Hunan Province, Hunan University, Changsha 410082, China; fangzhen2@hnu.edu.cn (Z.F.); xiaolin.meng@nottingham.ac.uk (X.M.)

[2]   Nottingham Geospatial Institute/Sino-UK Geospatial Engineering Centre, The University of Nottingham, Nottingham NG7 2TU, UK

[3]   Jiangsu Hydraulic Research Institute, Nanjing 210017, China; xieyilin-1983@163.com

[4]   College of Civil Engineering, Fuzhou University, Fuzhou 350116, China; fanqian@fzu.edu.cn

*   Correspondence: surveying@hnu.edu.cn

**Abstract:** Monitoring the dynamic responses of bridge structures has received considerable attention. It is important to synchronously measure both the quasi-static and dynamic displacements of bridge structures. However, the traditional accelerometer method cannot capture the quasi-static displacement component, although it can detect the dynamic displacement component. To this end, a novel composite instrument of a smartstation was proposed to monitor vibration displacements of footbridges. Full-scale experiments were conducted on a footbridge to validate the feasibility of the composite instrument-based monitoring method. A Chebyshev filter and wavelet algorithms were developed to process the composite instrument measurements. It was concluded that the measurement noise of the composite instrument was mainly distributed in a frequency range of 0–0.1 Hz. In two case studies with displacement peaks of 5.7–10.0 mm and 1.3– 2.5 mm, the composite instrument accurately identified the quasi-static and dynamic displacements. The composite instrument will be a potential tool for monitoring structural dynamics because of its enhanced overall performance.

**Keywords:** composite instrument of a smartstation; accelerometer; dynamic displacement; vibration frequency; structural health monitoring

## 1. Introduction

The measurement of three-dimensional quasi-static and dynamic displacements of bridge structures is an important task for monitoring their structural health [1]. Vibration displacements of bridge structures are usually measured by traditional methods such as the Global Navigation Satellite System (GNSS) and an accelerometer. GNSS technologies are capable of identifying dynamic displacement components (instantaneous deformation) at millimeter-level accuracy with a sampling rate up to 20 Hz, even 100 Hz [2,3]. However, when they are used to identify quasi-static displacement components induced by temperature changes or vehicle loads, their measurement accuracies are limited within a relatively low range from 10 to 20 mm because of multipath signal errors [4,5]. Moreover, although the accelerometer is capable of accurately detecting dynamic displacements of bridges by double integration of the accelerations, it cannot detect quasi-static displacements of bridges because of drift errors when the excitation term is relatively longer [6,7].

In order to solve the above-mentioned problem, a composite instrument of a smartstation was proposed to synchronously monitor both the dynamic and quasi-static displacements of bridge



structures. The composite instrument combines a high-performance robotic total station (RTS), which is also called terrestrial positioning system (TPS), and a powerful GNSS unit in one instrument. The GNSS unit is fully integrated into the RTS, rather than a simple combination of two sensors.

The greatest advantage of the composite instrument of the smartstation is that it can accurately capture not only three-dimensional coordinates of the monitored target, but also the Global Positioning System (GPS) time information with nanosecond-level accuracy. The GPS time information is very important for multi-sensor data fusion. A prototype of the smartstation system was designed and developed for the first time by Ingensand et al. in 1993. The invention involved a terrestrial surveying system comprising an electro-optic total station for the combined measurement of angle and distance, a connection, and a receiver for a satellite position-measuring system [8]. In order to evaluate the composite instrument of the smartstation, a case survey was conducted on the Porte Palatine, an archaeological building site in Turin. The composite instrument-based technique was considered to be a valuable solution for surveying cultural heritage [9]. The technique has been widely employed in many survey scenarios such as topography, boundary, construction site stakeout and utilities surveys [10].

The composite instrument was used to monitor the dynamic responses of a bridge structure in this study. A Leica TCA2003 RTS instrument has been used to monitor the structural dynamic responses of the Wilford Suspension Bridge in Nottingham, UK. However, very little movement of the bridge was measured due to the instrument's slow data sampling-rate of 1 Hz [11]. Ambient vibration measurements of the Bosphorus Suspension Bridge in Turkey were conducted using a TCA 2003 RTS instrument [12]. Seven vertical and lateral frequencies were detected from the RTS measurements, which were in the frequency range of 0–0.5 Hz. A new generation Leica 1201 RTS with a nominal sampling-rate of 10 Hz was used to monitor the dynamic responses of the Gorgopotamos Railway Bridge in Greece in response to passing trains [13,14]. The vertical displacements of the bridge with peaks of 2.5 to 6 mm, and its dominant frequencies, which were distributed in a frequency range of 3.18–3.63 Hz, were successfully identified [13,14]. The collocated sensors consisting of RTS, GNSS and an accelerometer were adopted to measure displacements in a stiff footbridge in Greece, and were found to be capable of measuring three-dimensional displacements with amplitudes of a few millimeters and identifying oscillation frequencies [15]. To overcome the RTS shortcoming of its low data sampling-rate, two approaches have been presented to increase the RTS sampling-rate from 10 Hz to 20 Hz to determine the vertical displacements of bridges [16]. It was concluded that the geodetic instruments, GNSS and RTS, in combination with accelerometers can be safely used to measure the semi-static and dynamic deflections of stiff bridges [17]. It is believed that the composite instrument of the smartstation will show better overall performance than the RTS because of its updated hardware and software [10].

The focus of this case study is on verifying the feasibility of using the composite instrument to monitor bridge dynamic responses. Background noise in three directions in composite instrument measurements were obtained and analyzed in both time and frequency domains for the purpose of characterizing the smartstation measurement noise. The composite instrument was used to monitor the dynamic responses of the studied footbridge, which were induced by a group of three people jumping, and a gentle wind and occasional pedestrians. The quasi-static and dynamic displacements of the footbridge were accurately detected by the composite instrument.

## 2. Instrumentation and the Studied Footbridge

### 2.1. Composite Instrument of the Smartstation

The composite instrument of the smartstation (Figure 1) was employed to monitor the dynamic deformation of the footbridge used in this study. The composite instrument combines a high-performance Leica TS30 robotic total station (RTS), which is also called a terrestrial positioning system, TPS), and a powerful Leica ATX1230+ GNSS smart-antenna in one instrument. The GNSS unit

is fully integrated into the RTS. All data are stored in the same database on the same CompactFlash card, all GNSS and RTS operations are controlled via the RTK keyboard with the entire software in the RTS, and all measurements, status and other information are displayed on the RTS screen (Figure 1).

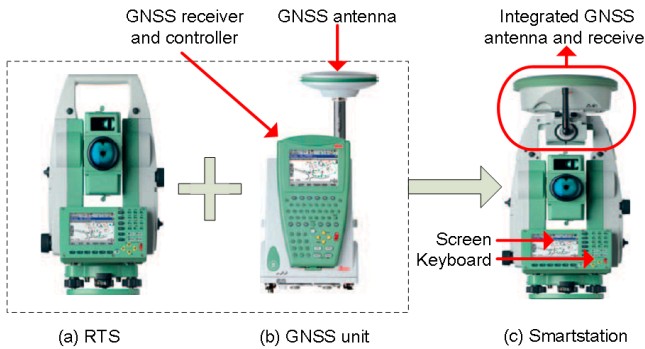

**Figure 1.** The composite instrument of the smartstation. The composite instrument combines a high-performance Leica TS30 robotic total station (RTS) (**a**), a powerful Leica GNSS unit (**b**) in one instrument and integrated GNSS antenna and receiver (**c**).

The composite instrument has the capability of automatic target recognition (ATR), which allows fast, dynamic tracking of targets in three dimensions. The composite instrument can obtain three-dimensional coordinates of a prism with a true sample-rate of up to 5–7 Hz, as well as the nanosecond-level GPS time information in each record. The nominal sampling-rate of the composite instrument of the smartstation is 10 Hz. However, the true sampling-rate of the instrument is 5–7 Hz became of missing data. The composite instrument is one of the best performing coordinate measurement instruments, with a nominal angle accuracy of ± 0.5″ and a nominal distance accuracy of 0.6 mm + 1 ppm.

*2.2. Accelerometer and Precise Time Data Logger*

A Kistler 8392A2 triaxial accelerometer with a data sample-rate of 150 Hz was used to monitor the bridge dynamic responses to verify the results from the composite instrument. It is ideal for modal tests of large structures, with technical parameters in the acceleration range of 2 g, a sensitivity of 500 mV/g, resolution of 0.3 mg and a weight of 40 g. For the purpose of solving the problem of time synchronization, a precise time data logger (PTDL) was applied to record the accelerator data (Figure 2c). It can tag the GPS time onto the external data from the accelerometer because it contains a built-in, low-cost GPS chip and a small antenna [18]. Thus, the data from both the composite instrument and the accelerator can be applied for data fusion or comparison based on the same GPS time system.

*2.3. Description of the Nottingham Wilford Suspension Bridge*

Full-scale experiments were performed on the Wilford Suspension Bridge in Nottingham, UK (Figure 2a). The bridge, which is also known as the Meadows Suspension Bridge, is a combined suspension pedestrian footbridge and aqueduct that crosses the River Trent, linking the town of West Bridgford to the Meadows. The historical bridge was constructed in 1904 with a 69.0-m-length main span. Following a restoration in 1983, it was closed to pedestrians in 2008 for a major restoration because of debris falling from the bridge deck. In the last two decades, the bridge has been utilized as the testbed for the purpose of monitoring research with innovative sensors and approaches. The first three modal frequencies of the bridge, i.e., 1.44 Hz, 2.79 Hz and 4.66 Hz, have been computed with its finite element model in previous research [19]. The modal parameters of the bridge might have changed due to the major restoration in 2008.

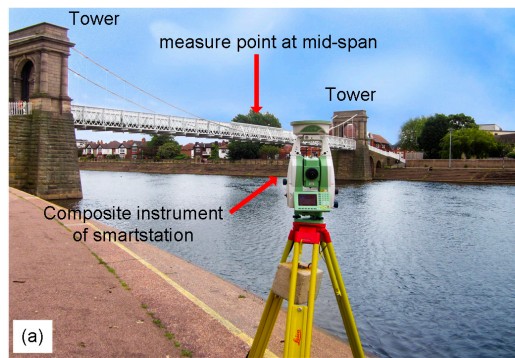 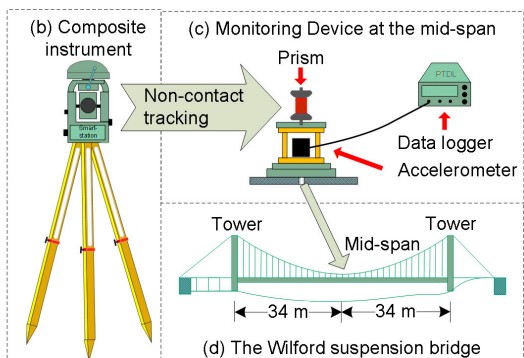

**Figure 2.** The Wilford Suspension Bridge in Nottingham and the instrumentation. The composite instrument was set near the bridge to monitor the structural dynamic response (**a**); the composite instrument (**b**) in non-contact mode locked and tracked a 360° prism that was installed outside a cage with a triaxial accelerometer, connected to a precise time data logger (**c**); the monitoring devices were fixed at the mid-span of the footbridge for dynamic response monitoring (**d**).

## 3. Methodology for the Field Measurements

Three experiments with different working conditions were performed in this study (Table 1). Before the full-scale experiments were undertaken, a static experiment (Case 1) was carried out on the campus of the University of Nottingham to understand the noise characteristics in the measurements of the composite instrument. The two full-scale experiments were conducted on the Wilford suspension bridge in Nottingham to verify the measurement of the vibration displacement of footbridges using the composite instrument (Figure 2). In case 2 and 3, the vibration displacement of the footbridge was monitored by the composite instrument, with the displacement being induced by three people jumping, and gentle wind and an occasional pedestrian (Case 3). The vibration displacements cover both the quasi-static and dynamic displacement components.

**Table 1.** The details of the three experimental cases.

| Case | GPS Time | Description | Instrumentation |
|---|---|---|---|
| 1 | From second 314,100 to 314,700, GPS week 1768 | Measuring background noise in measurements of the composite instrument | A composite instrument of the smartstation |
| 2 | From second 312,500 to 313,700, GPS week 1698 | Measuring three people jumping -induced vibrations of the footbridge | A composite instrument and a tri-axial accelerometer |
| 3 | From second 476,100 to 477,300, GPS week 1699 | Measuring gentle wind and occasional pedestrian-induced vibrations of the footbridge | A composite instrument and a tri-axial accelerometer |

In the static experiment (Case 1), the composite instrument and the prism were each mounted on a tripod. The two tripods were setup on two stable sites on the campus of the University of Nottingham, with a distance of 60 m between the two tripods. The 3D coordinates of the prism were measured and recorded continuously by the composite instrument with a true sample-rate of 5~7 Hz. A three-axis coordinate system was constructed with the *x*-axis aligned with the measuring direction from the composite instrument to the prism, and the *z*-axis was coincident with the gravity direction. Because the prism is not moving, any displacements found in the composite instrument observations can be regarded as background noise [20,21]. We acquired the background noise in three directions over about 60 min. We graphically analyzed a 600-s-long displacement time series from 314,100 s to 314,700 s, GPS week 1768.

In the full-scale experiment (Case 2), the composite instrument station was set up on stable site A2, located at the downstream side of the footbridge, 67.2 m away from the measuring point at the mid-span of the footbridge (Figure 3). The composite instrument acquired vibration displacement at the measuring point with a true sample-rate of 5–7 Hz. The backsight point of the composite instrument was set at the upstream (north) side of the studied bridge, approximately 74.8 m away from the

corresponding instrument station. It is necessary to choose the right site as the backsight point because of the need for a line-of-sight between the backsight point and the instrument station. The measuring point was set at the midspan of the footbridge. Monitoring devices were installed at the measuring point of the midspan, including a Leica GRZ122 360° prism, a Kistler 8392A2 triaxial accelerometer, a PTDL data logger produced by the research team at the University of Nottingham, and a special cage. The special cage was fixed on the rail of the midspan of the footbridge. The accelerometer was fixed inside the cage whereas the prism was installed outside the cage (Figure 2c). The PTDL data logger was adopted to record the accelerometer data in the GPS time system, with a data sampling-rate of 100 Hz.

The forced vibrations of the footbridge were monitored by both the composite instrument and accelerometer sensors, which were induced by three people jumping simultaneously at the midspan for several minutes. We acquired both displacement and acceleration time series from 312,500 s to 313,700 s, GPS week 1698.

In the full-scale experiment (Case 3), the instrument station was located at the A3 site, 64.4 m away from the measuring point of the footbridge (Figure 3). Similar to the previous case, the vibration responses of the footbridge were measured by both the composite instrument and accelerometer sensors, which were induced by a gentle wind and an occasional pedestrian. We synchronously acquired one set of 1200-s-long displacement time series with a data sampling-rate of 5-7 Hz, and one set of 1200-s-long acceleration time series with a data sampling-rate of 100 Hz, from 476,100 s to 477,300 s, GPS week 1699.

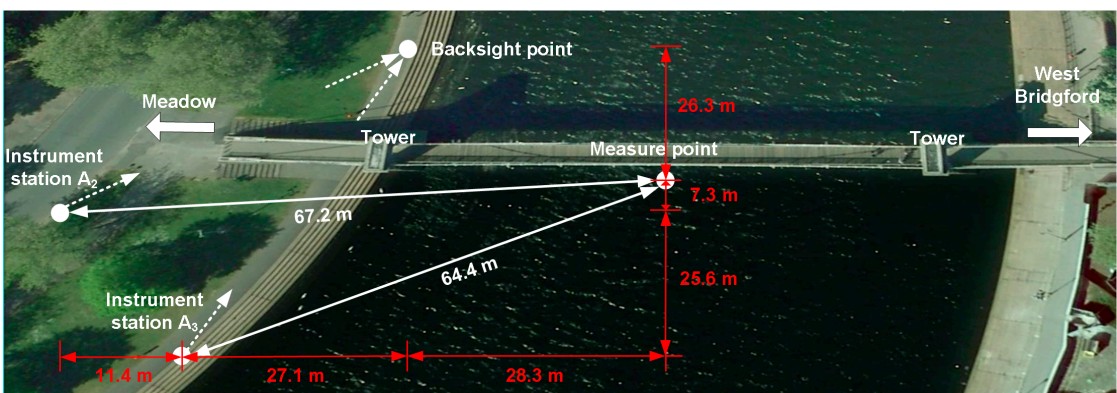

**Figure 3.** Instrument layout of full-scale experiments performed on the Wilford Suspension Bridge. The instrument stations for the composite instrument were set at the A2 and A3 sites in Cases 2 and 3, respectively. The backsight point was set at the upstream (north) side of the bridge; the measuring point was set at the mid-span of the studied footbridge.

## 4. Results and Discussion

The background noise in the measurements of the composite instrument were detailed in both time and frequency domains by the continuous Morlet wavelet transform (CMWT) algorithm. The dynamic responses of the footbridge were induced by a group of three people jumping, and gentle wind and the occasional pedestrian. The two cases were monitored and identified by the composite instrument.

### 4.1. Background Noise in the Composite-Instrument Measurements

In order to characterize the measurement noise of the composite instrument in detail, a set of displacement time series in three directions were acquired by the composite instrument in the static experiment of Case 1. The displacement in each record included the point number, GPS time, x-coordinate, y-coordinate and z-coordinate. A standardized identification procedure was used to preliminarily process these original records for the purpose of removing duplicates and outliers, recovering missing records, and so on [21]. Firstly, the duplicate records with an identical recording time were removed from the original records; these were caused by the high data-recording rate of the composite instrument. Secondly, the outliers were removed from the original displacement records

according to a threshold of three times the standard deviation. Finally, the missing records were recovered by a linear interpolation method according to a time interval of 0.1 s between two adjacent records. The MATLAB codes were developed to preliminarily process those original records with the above methods by the authors.

Figure 4 graphically depicts the profile of the background noise in the measurements of the composite instrument in three directions when the composite instrument is 60 m away from the prism. The displacement in the *x*-axis are primarily caused by the *x*-axis being parallel with the sighting-distance direction, which shows the resolution of the distance meter. The y- and z-axes noise is caused by both angle and distance errors because they are determined by the distance and their corresponding angles. The amplitude of the background noise fluctuates between −1.4 mm and 0.6 mm in the *x*-axis, between −0.9 mm and 1.0 mm in the *y*-axis, and between −1.1 mm and 0.8 mm in the *z*-axis (vertical direction). In the horizontal plane, the *x*-axis is coincident with the sighting line of the composite instrument whereas the *y*-axis is perpendicular to the sighting line. All standard deviations (SD) and mean absolute errors (MAE) of the background noise in each direction are less than 0.5 mm (Table 2). The results show that the measurement accuracies of the composite instrument monitoring technique meet the requirements for monitoring the dynamic deformation of footbridges.

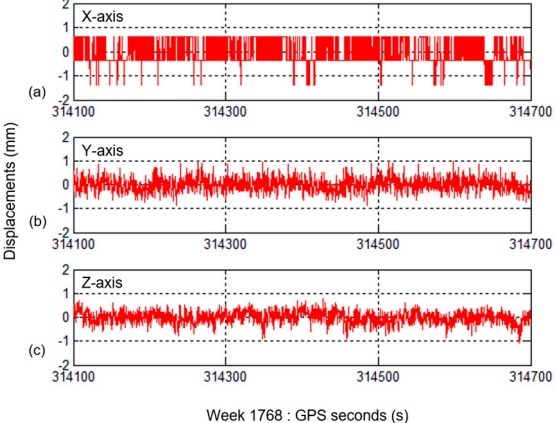

**Figure 4.** **(a–c)**Time history of the background noise in the composite instrument measurements in experimental Case 1. The prism was set up on a stable site, 60 m away from the composite instrument.

**Table 2.** Standard deviations (SD) and mean absolute errors (MAE) of background noise in the composite instrument measurements.

| Direction | SD (mm) | MAE (mm) |
| --- | --- | --- |
| *x*-axis | 0.5 | 0.4 |
| *y*- axis | 0.3 | 0.2 |
| *z*- axis | 0.2 | 0.2 |

Figure 5 graphically depicts the frequency domain characteristics of the composite instrument measurement noise, which correspond to the displacements shown in Figure 4. The spectra of background noise in three directions were produced using the CMWT algorithm (seeing details in references [22,23]).

It is noted that the noise is mainly distributed in a frequency range of no more than 0.1 Hz, and is caused by the angle and distance measurement errors of the composite instrument. The noise energy in a frequency range of more than 0.1 Hz is relatively low, and is mainly caused by instrument self-noise such as Gaussian noise and white noise [24]. Besides, it is obvious that the noise energy in the *z*-axis is lower than that in the x- and y-axes. This characteristic agrees well with the SD and MAE values for the background noise in the composite instrument measurements, as shown in Table 2. The accuracies of the horizontal displacements are determined by the horizontal-angle and distance

measurements, and the accuracies of vertical displacements are determined by the vertical-angle and distance measurements. Horizontal angle accuracies are usually lower than vertical angle accuracies because the horizontal angle measurements need a horizontal orientation angle. This results in the displacement measurement accuracies in the *z*-axis direction (vertical direction) being relatively higher than those in the other directions.

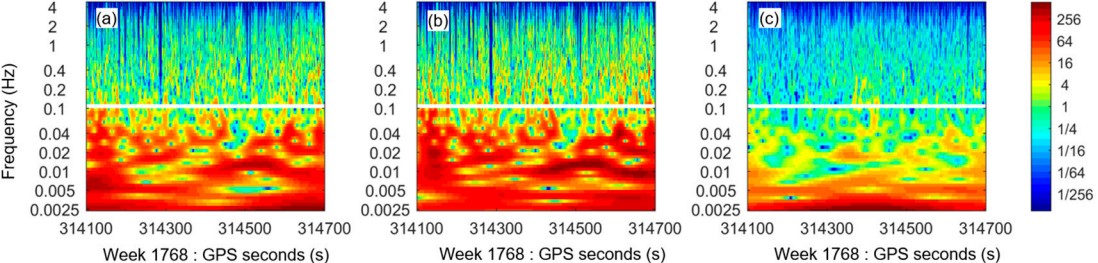

**Figure 5.** The continuous Morlet wavelet transform (CMWT)-based spectra of the measurement noise of the composite instrument in the *x*-axis (**a**), *y*-axis (**b**) and *z*-axis (**c**), corresponding to displacement time series in Figure 4 with white lines indicating the frequency of 0.1 Hz. The horizontal axis indicates the GPS time, which is composed of GPS weeks and week seconds. The color pattern indicates the amplitudes of the energy spectrum that are dimensionless.

### 4.2. Three People Jumping-Induced Vibrations

In experimental Case 2, both the composite instrument and accelerometer sensors were used to monitor the dynamic responses of the Wilford Suspension Bridge, which were induced by a group of three people synchronously jumping for around 10 s each time on its midspan. A set of 1200 s displacement time series with an interval of 0.1 s were acquired by the composite instrument. Also, a set of 1200 s acceleration time series were acquired with an interval of 0.01 s by the accelerometer in the same GPS time frame. Using codes written by the authors, these two groups of time series were preliminarily processed to remove duplicates and outliers, and recover missing records, as in the previous case.

Figure 6a,b graphically depict the overall vibration displacements of the footbridge in the longitudinal and lateral directions measured by the composite instrument. Although the vibration displacements of the footbridge are very small in both the longitudinal and lateral directions, they are accurately measured by the composite instrument. The longitudinal displacements change between −1.3 mm and 1.0 mm with a SD of 0.3 mm. The lateral displacements change between −1.7 mm and 1.6 mm, with a SD of 0.5 mm. Figure 6c graphically depicts the vibration displacements of the footbridge in the vertical direction, measured by the composite instrument. Five significant displacement peaks in the vertical direction correspond to the five times that three people jumped at the midspan of the footbridge.

The smartstation-measured vibration displacement of the footbridge were separated into two components, i.e., quasi-static displacements and dynamic displacements using the Chebyshev high pass filter [25,26]. Also, the dynamic displacements of the footbridge (see right column of Figure 6) were computed from the accelerometer data by double integration [6,27], and these were used to validate the composite instrument results. It is known that the quasi-static displacements of the footbridge cannot be derived from the accelerometer data because of the trend errors [28].

Although all quasi-static displacements of the footbridge in three directions have low amplitudes of about 1 mm (see the black curves in the left column of Figure 6), the composite instrument can accurately detect these quasi-static displacement components. This suggests that measurement errors of the composite instrument should be lower than a millimeter. The curve of the vertical quasi-static displacements shows a strong correlation between the vertical deflections and experimenter loads (Figure 6c). When three people walked onto the bridge in the first tens of seconds of the experiment, the curve of the vertical quasi-static displacements shows a gradual downward tendency. When three

people stay on the midspan of the footbridge, the vertical quasi-static displacements fluctuate close to one millimeter. when three people walk off the footbridge at the end of the experiment, the vertical quasi-static displacements recover to a zero value.

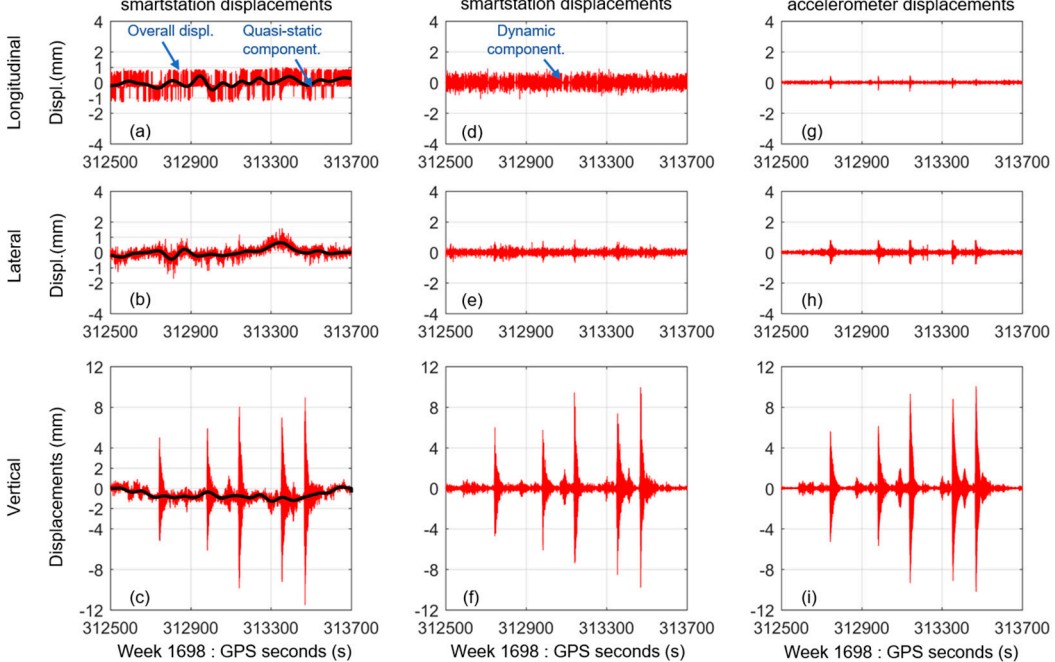

**Figure 6.** Displacement time series of the footbridge in the longitudinal, lateral and vertical direction induced by a group of three people jumping in experimental Case 2. The vibration displacements of the footbridge measured by the composite instrument (the red curves in the left column) were separated into two parts, i.e., the quasi-static displacements (the black curves in the left column) (**a**–**c**); the dynamic displacements (central column) (**d**–**f**). The dynamic displacements were also computed from the accelerometer measurements by double integration (right column) (**g**–**i**).

Two sets of dynamic displacements of the footbridge were derived from the composite instrument and the accelerometer data (see the center and right columns of Figure 6). It is obvious that they are similar to each other in each direction. In both the longitudinal and lateral directions, the dynamic displacements have small amplitudes of less than 0.5 mm. The measurement errors of the composite instrument should be lower than a half-millimeter. The dynamic displacements computed from the accelerometer data have relatively lower amplitudes than the ones computed from the composite instrument data. The main reason for this is that the random noise of accelerometer data is partly eliminated when the displacements are computed from the accelerations.

In the vertical direction, both two sets of dynamic displacements as shown in Figure 6f,i depict five significant peaks, which correspond to the five times that three people jumped on the bridge. The corresponding peak differences between these two sets of dynamic displacements are less than 1.0 mm and the peak difference ratios are no more than 10.0% (Table 3). This shows that the composite instrument can accurately measure dynamic displacements of the footbridge with peak values of several millimeters (from 5.7 mm to 10.0 mm).

Figure 7 depicts three pairs of wavelet-based spectra for the overall vibration displacements and dynamic displacements derived from the composite instrument data, and the dynamic displacements derived from the accelerometer measurements in the vertical direction. These wavelet-based spectra provide the local characteristics for the displacement time series. Each three-dimensional spectrum shows five significant displacement peaks, corresponding to the five times that three people jumped at the midspan of the footbridge. The three-dimensional spectrum of the overall vibration displacements shows significant differences with the other two three-dimensional spectra of the dynamic displacements.

The reason is that the overall vibration displacements covers relatively higher random noise and quasi-static displacement components. This is similar to the two spectra of dynamic displacements derived from the composite instrument and accelerometer data.

**Table 3.** Comparison of peak values of two sets of dynamic displacement in the vertical direction derived from the composite instrument and accelerometer measurements in Case 2.

| Event | Peak Displacements (mm) | | Diff. (mm) (①−②) | Ratio (%) (|①−②|)/② |
|---|---|---|---|---|
| | Composite Instrument① | Accelerometer② | | |
| 1 | 6.0 | 5.6 | 0.4 | 7.1 |
| 2 | 5.7 | 6.1 | −0.4 | 6.6 |
| 3 | 9.5 | 9.3 | 0.2 | 2.2 |
| 4 | 7.7 | 8.6 | −0.9 | 10.5 |
| 5 | 10.0 | 10.1 | −0.1 | 1.0 |

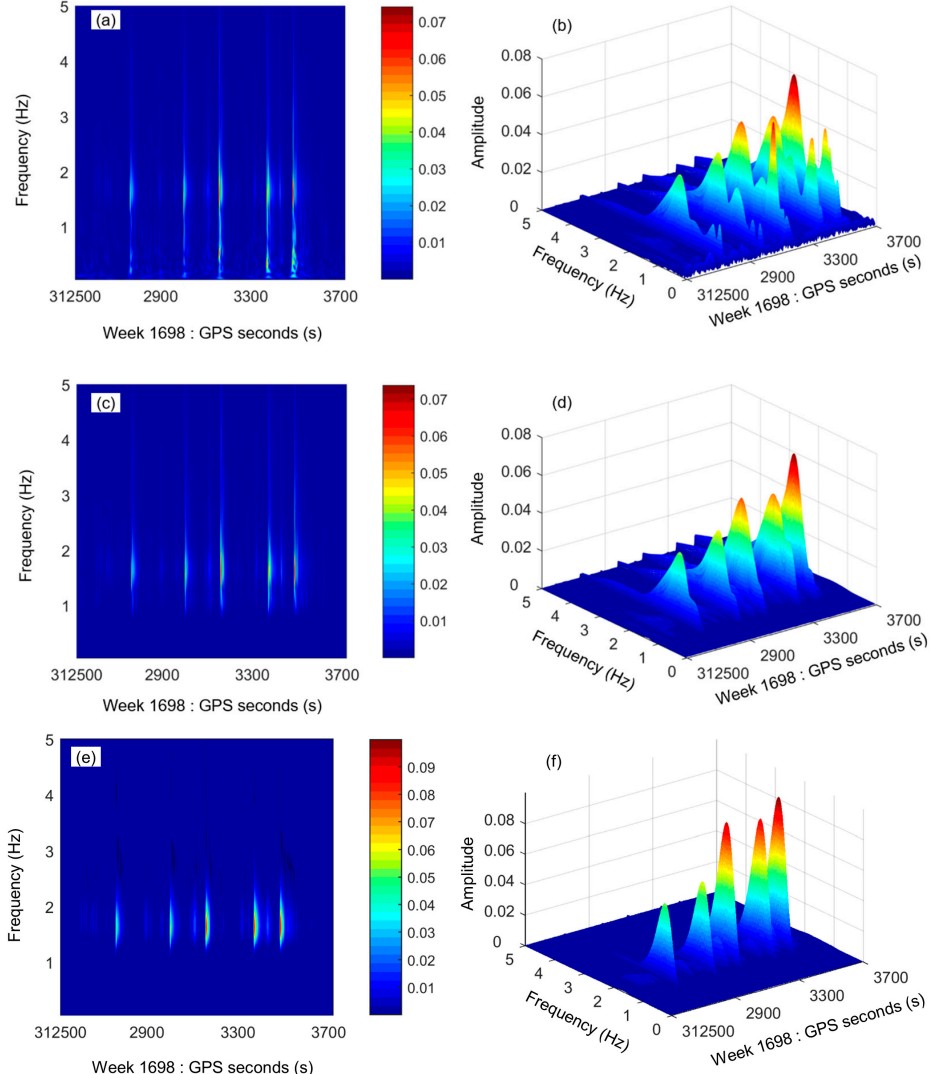

**Figure 7.** Wavelet-based spectra of the displacement time series in the vertical direction: two- and three-dimensional spectra of overall vibration displacements (**a**,**b**) and dynamic displacements (**c**,**d**) derived from the composite instrument data, and dynamic displacements (**e**,**f**) derived from the accelerometer data. The horizontal axis indicates the GPS time, which is composed of GPS weeks and week seconds. The color pattern indicates the amplitudes of the energy spectrum that are dimensionless.

### 4.3. Wind and Occasional Pedestrian-Induced Vibrations

The vibration responses of the footbridge, which were induced by gentle wind and occasional pedestrians, were monitored in experimental Case 3. We synchronously acquired 1200-s vibration displacements of the footbridge using the composite instrument, and 1200-s accelerations using the accelerometer in the same GPS time frame. The composite instrument acquired data with a true sampling-rate of 5–7 Hz whereas the accelerometer acquired data with a sample-rate of 100 Hz.

Figure 8a–c. graphically depict the overall vibration displacements of the footbridge in three directions measured by the composite instrument. They were preliminarily processed to remove duplicates and outliers, and recover missing records as previously outlined above. All of these displacements in three directions have small amplitudes with maximal amplitudes of 1.3 mm, 2.1 mm, and 2.5 mm in the longitudinal, lateral and vertical directions, respectively. It is obvious that the composite instrument can sensitively identify the vibration displacements even though the vibration displacements only have maximal amplitudes of about 2 mm.

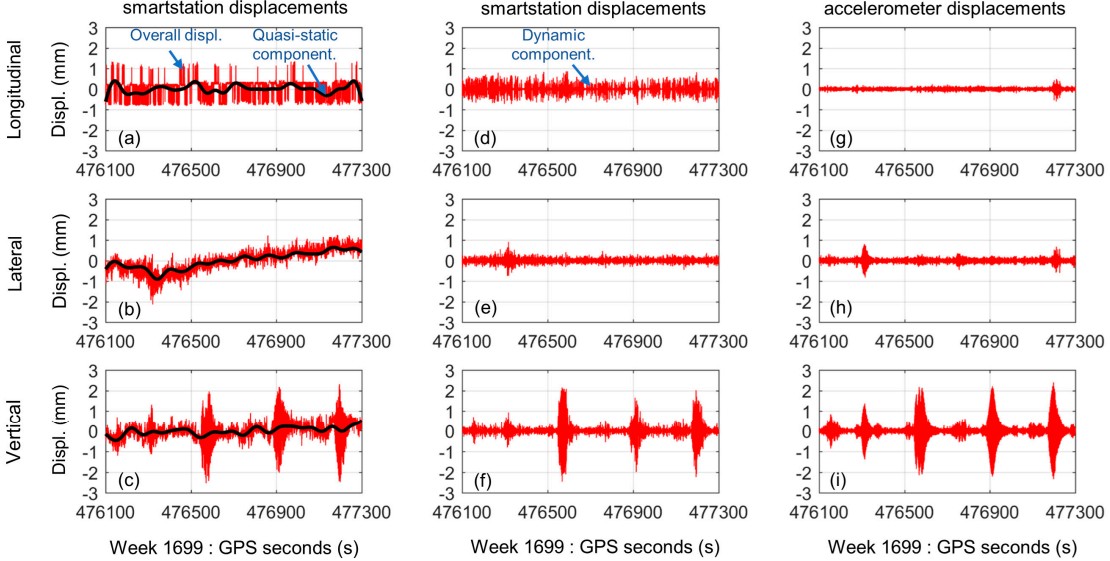

**Figure 8.** Time histories of displacements of the footbridge in three directions induced by gentle wind and the occasional pedestrian in experimental Case 3. The overall vibration displacements measured by the composite instrument (the red curves in the left column) were separated into two parts, i.e., the quasi-static displacements (the black curves in the left column) (**a**–**c**). Dynamic displacements (central column) (**d**–**f**). The dynamic displacements (right column) were also computed from the accelerometer data by double integration (**g**–**i**).

Quasi-static displacements of the footbridge were also identified from the composite instrument data, which are depicted by the black curves in the left column of Figure 8. They fluctuate within an amplitude of 1 mm, which is usually caused by thermal expansion, wind, pedestrian-weight and so on. Two sets of dynamic displacements of the footbridge in three directions were derived from both the composite instrument and accelerometer data, which are depicted as shown in the central and right columns of Figure 8, respectively. The dynamic displacements in each direction that were derived from the composite instrument and accelerometer data are similar to each other. In the vertical direction, the three peak values of the dynamic displacements derived from the composite instrument data are about 2 mm, which agree well with those for the dynamic displacements derived from the accelerometer data. This confirmed once again that the composite instrument can measure dynamic displacements with maximum amplitudes of about 2 mm.

The modal frequencies of the footbridge in the vertical direction were identified using the fast Fourier transform (FFT) method (Figure 9). Modal frequencies of 1.677 Hz and 2.885 Hz were detected from the dynamic displacement time series of the composite instrument in the vertical direction. Also,

modal frequencies of 1.687 Hz and 2.875 Hz were detected from the dynamic displacement time series of the accelerometer in the vertical direction. The difference in the values of the frequencies detected from both the composite instrument and accelerometer are no more than 0.01 Hz, and the difference ratio are less than 0.6%. This shows that the composite instrument is capable of detecting relatively low vibration frequencies of the footbridge.

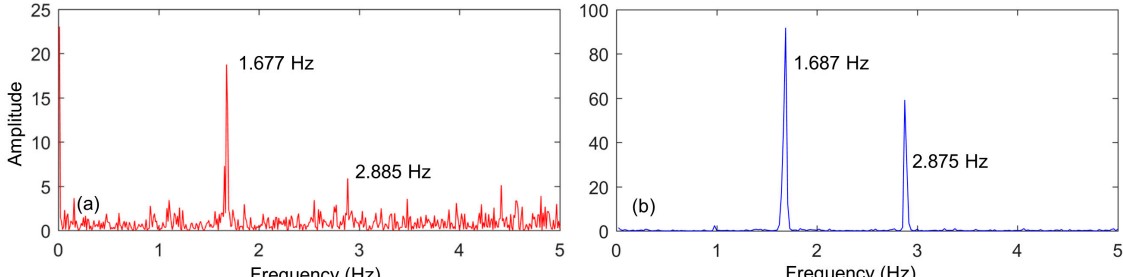

**Figure 9.** Comparison of FFT-based spectra of two dynamic displacements in the vertical direction: (**a**) spectrum of the dynamic displacements derived from the composite instrument data, and (**b**) the spectrum computed from the accelerometer data.

The distribution ranges of the longitudinal and lateral displacements of the measuring point of the footbridge are graphically depicted in Figure 10. These displacements were derived from the composite instrument data in experimental Case 2 and 3. In each case, the distribution range of the overall vibration displacements in the lateral direction are larger than the one in the longitudinal direction. The distribution range of the quasi-static displacements show similar characteristics. The reason is that the wind along the lateral directions causes relatively larger vibrations in the lateral direction, thus, the tendency for the increased displacements along the wind direction is evident.

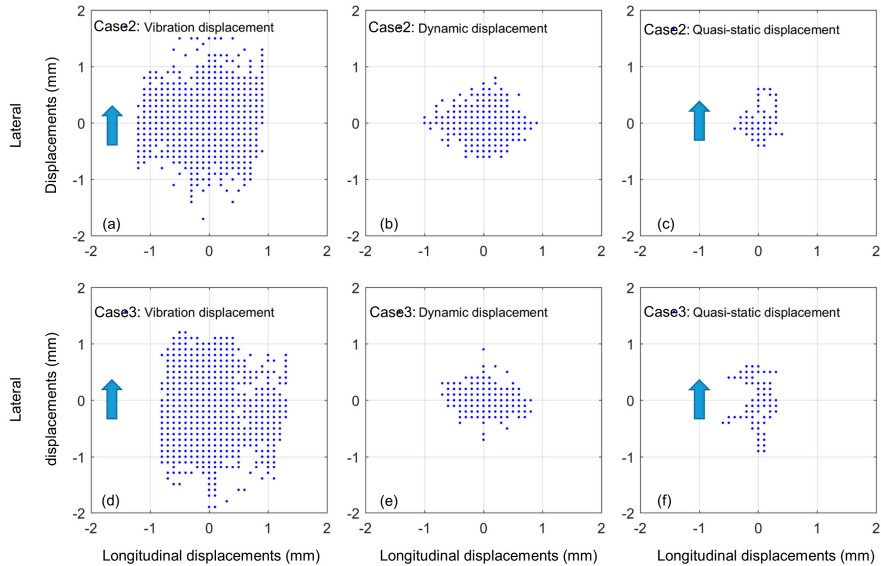

**Figure 10.** The distribution ranges of the longitudinal and lateral displacements of the measuring point of the studied footbridge, derived from the composite instrument measurements. The overall vibration displacements (**a**), dynamic displacements (**b**) and quasi-static displacements (**c**) in Case 2, and the corresponding ones (**d**–**f**) in Case 3. The distribution range of the overall vibration displacements in the lateral direction are significantly larger than the ones in the longitudinal direction as shown in subfigure (**a**). This is because the direction of the wind is along the lateral direction of the bridge, with blue arrows indicating the direction of the wind. Similar results can be found in subfigures (**c**,**d**,**f**).

## 5. Conclusions

A revolutionary surveying system using a composite instrument was proposed to monitor three-dimensional vibration displacements of a footbridge. Not only is it capable of detecting quasi-static and dynamic displacements of the footbridge, but it can also obtain nanosecond-level time information for each record, which is valuable for the fusion of multi-sensor data fusion. A CMWT algorithm was developed to understand the noise characteristics in the composite instrument measurements in the frequency domain. Experiments were conducted on the Nottingham Wilford Suspension Bridge to verify the measurements of vibration displacements of footbridges using the composite instrument. The Chebyshev highpass filter was adopted to separate quasi-static and dynamic displacements from the composite instrument measurements. Both wavelet and FFT algorithms were employed in the spectral analyses of the displacement time series in the frequency domain.

Firstly, we verified that the composite instrument possesses the capacity to accurately identify both quasi-static and dynamic displacements of footbridges with several millimeter amplitudes. The dynamic displacements created by three people jumping on the footbridge were identified by the composite instrument, with displacement peak values from 5.7 to 10 mm. The accuracy of the measurement was superior to 1 mm. The quasi-static displacement components of the footbridge were identified from the composite instrument data with submillimeter-level accuracy. The corresponding peak differences between two sets of dynamic displacements derived from the composite instrument and accelerometer data are less than 1.0 mm and the peak difference ratios were no more than 10.0%. In other case study, slight wind and occasional pedestrian-induced vibration displacements of the footbridge, with peak displacements from 1.3 to 2.5 mm, were accurately identified by the composite instrument. Relatively low vibration frequencies of 1.677 Hz and 2.885 Hz in the footbridge in the vertical direction, were identified from the composite instrument measurements.

The second contribution of this study is that the composite instrument measurement noise was characterized in detail by a CMWT algorithm in the frequency domain. All of the standard deviations of the background noise in each direction are no more than 0.5 mm when the composite instrument is 60 m away from the prism. The main noise energy is distributed within a relatively low frequency range of no more than 0.1 Hz.

The composite instrument could potentially be used as a tool for synchronously detecting the quasi-static and dynamic displacements of bridge structures with satisfactory accuracy. The study only concentrates on displacement monitoring of footbridges, but the composite instrument could be also used to identify the modal frequencies of footbridges, which would significantly improve its overall performance and data acquisition algorithm.

**Author Contributions:** Conceptualization, J.Y. and X.M.; methodology, J.Y.; software, J.Y.; validation, J.Y., Y.X. and Q.F.; formal analysis, J.Y.; investigation, J.Y., Q.F. and Z.F.; resources, J.Y.; data curation, J.Y.; writing—original draft preparation, J.Y.; writing—review and editing, Z.F.; visualization, J.Y.; supervision, X.M.; project administration, J.Y., X.M.; funding acquisition, J.Y. All authors have read and agreed to the published version of the manuscript.

**Funding:** This study is supported by the National Key R&D Program of China (No. 2016YFC0800207) and Changsha Science and Technology Project (No. kq1907110).

**Acknowledgments:** The University of Nottingham is thanked for supplying instruments in the field experiments.

**Conflicts of Interest:** The authors declare no conflict of interest.

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
