# Peer review of "Measurement of Quasi-Static and Dynamic Displacements of Footbridges Using the Composite Instrument of a Smartstation and an Accelerometer: Case Studies"

_remotesensing, doi:10.3390/rs12162635_

Round 1

Reviewer 1 Report

The paper is based on appropriate methods (to some extent), which are typically used for the measurements of the bridges. However, this paper contains very little new or useful scientific results and might not appear to warrant publication in a good journal. Peer-reviewed manuscripts must be encouraged, and this paper for sure deserves a publication, but I strongly recommend that the author should publish this paper somewhere else or in conference proceedings or in other journal. A paper published in a journal indexed in JCR shall provide new methods or new results, and this article appears to be rather an academic technical report than a research article. This type of analysis is carried out by students as part of engineering studies. The authors used a total station with an integrated GNSS antenna and used it to analyze bridge displacements, which is carried out by companies dealing with building and structures monitoring.

I do not see anything scientific in this work, it does not bring any added value to the field, therefore I do not recommend this article for publication.

Reviewer 2 Report

It‘s a very interesting article in the case that for many years testing the properties of bridges we tried to determine the reliability of the data in the low frequency bands up to 0.3 Hz. The question arises, why you do not provide accelerometer parameters (such as frequency band, sensitivity and mass)? Therefore usually the biggest problem with accelerometers is reliability in the low frequency range.

The key question is, does whether the results of the accelerometers used in the low frequency bands are reliable  and have the accelerometers been calibrated in the frequency band to 0.1 Hz?

This aspect should be described in more detail.

And why did you choose an accelerometer and an object bridge for comparison, rather than a more reliable calibration system for a vibrator with a displacement measurement system?

Author Response

Please see the attchement.

Reviewer 3 Report

An interesting paper. Some minor comments and suggestions is in the attached file.

There are some minor issues which need to be addressed:

1) Explain the meaning of time-frequency domain. These two are usually mutually exclusive.

2) Standard deviation in always positive (+- should not be used here).

3) Some signals are quite unusual (see Fig. 4a), please explain.

4) explain the meaning and method behind figures 5, 7.

5) Sampling rate is 5-7, 10 or 100 Hz. This is not clear.

6) Fig. 10 is not clear, please explain better.

Author Response

Please seee the attachment.

Round 2

Reviewer 1 Report

I accept the text in a current form.

Reviewer 2 Report

Thank you for a good level article.